# PowerQuant: Automorphism Search for Non-Uniform Quantization

**Edouard Yvinec**[1,2] , **Arnaud Dapogny**[2] , **Matthieu Cord**[1] , **Kevin Bailly**[1,2]
Sorbonne Université[1], CNRS, ISIR, f-75005, 4 Place Jussieu 75005 Paris, France
Datakalab[2], 114 boulevard Malesherbes, 75017 Paris, France
ey@datakalab.com

## Abstract

Deep neural networks (DNNs) are nowadays ubiquitous in many domains such as computer vision. However, due to their high latency, the deployment of DNNs hinges on the development of compression techniques such as quantization which consists in lowering the number of bits used to encode the weights and activations. Growing concerns for privacy and security have motivated the development of data-free techniques, at the expanse of accuracy. In this paper, we identity the uniformity of the quantization operator as a limitation of existing approaches, and propose a data-free non-uniform method. More specifically, we argue that to be readily usable without dedicated hardware and implementation, non-uniform quantization shall not change the nature of the mathematical operations performed by the DNN. This leads to search among the continuous automorphisms of $(\mathbb{R}_+^*, \times)$, which boils down to the power functions defined by their exponent. To find this parameter, we propose to optimize the reconstruction error of each layer: in particular, we show that this procedure is locally convex and admits a unique solution. At inference time, we show that our approach, dubbed PowerQuant, only require simple modifications in the quantized DNN activation functions. As such, with only negligible overhead, it significantly outperforms existing methods in a variety of configurations.

## 1 Introduction

Deep neural networks (DNNs) tremendously improved algorithmic solutions for a wide range of tasks. In particular, in computer vision, these achievements come at a consequent price, as DNNs deployment bares a great energetic price. Consequently, the generalization of their usage hinges on the development of compression strategies. Quantization is one of the most promising such technique, that consists in reducing the number of bits needed to encode the DNN weights and/or activations, thus limiting the cost of data processing on a computing device.

Existing DNN quantization techniques, for computer vision tasks, are numerous and can be distinguished by their constraints. One such constraint is data usage, as introduced in Nagel et al. (2019), and is based upon the importance of data privacy and security concerns. Data-free approaches such as Banner et al. (2019); Cai et al. (2020); Choukroun et al. (2019); Fang et al. (2020); Garg et al. (2021); Zhao et al. (2019); Nagel et al. (2019), exploit heuristics and weight properties in order to perform the most efficient weight quantization without having access to the training data. As compared to data-driven methods, the aforementioned techniques are more convenient to use but usually come with higher accuracy loss at equivalent compression rates. Data-driven methods performance offer an upper bound on what can be expected from data-free approaches and in this work, we aim at further narrowing the gap between these methods.

To achieve this goal, we propose to leverage a second aspect of quantization: uniformity. For simplicity reasons, most quantization techniques such as Nagel et al. (2019); Zhao et al. (2019); Cong et al. (2022) perform *uniform* quantization, *i.e.* they consist in mapping floating point values to an evenly spread, discrete space. However, non-uniform quantization can theoretically provide a closer fit to the network weight distributions, thus better preserving the network accuracy. Previous work on non-uniform quantization either focused on the search of binary codes (Banner et al., 2019;

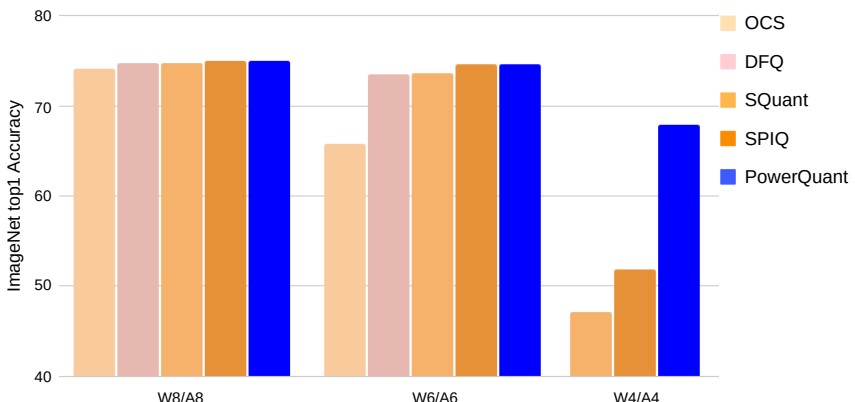

Figure 1: Comparison of the proposed method to other data-free quantization schemes on DenseNet 121 pre-trained on ImageNet. The proposed method (right bin in blue) drastically improves upon the existing data-free methods especially in the challenging W4/A4 quantization.

Hubara et al., 2016; Jeon et al., 2020; Wu et al., 2016; Zhang et al., 2018) or leverage logarithmic distribution (Miyashita et al., 2016; Zhou et al., 2017). However, these approaches map floating point multiplications operations to other operations that are hard to leverage on current hardware (e.g. bit-shift) as opposed to uniform quantization which maps floating point multiplications to integer multiplications (Gholami et al., 2021; Zhou et al., 2016). To circumvent this limitation and reach a tighter fit between the quantized and original weight distributions, in this work, we propose to search for the best possible quantization operator that preserves the nature of the mathematical operations. We show that this search boils down to the space defined by the continuous automorphisms of $(\mathbb{R}_+^*, \times)$, which is limited to power functions defined by their exponent. We optimize the value of this parameter by minimizing the error introduced by quantization. This allows us to reach superior accuracy, as illustrated in Fig 1. To sum it up, our contributions are:

- We search for the best quantization operator that do not change the nature of the mathematical operations performed by the DNN, *i.e.* the automorphisms of $(\mathbb{R}_+^*, \times)$. We show that this search can be narrowed down to finding the best exponent for power functions.

- We find the optimal exponent parameter to more closely fit the original weight distribution compared with existing (e.g. uniform and logarithmic) baselines. To do so, we propose to optimize the quantization reconstruction error. We show that this problem is locally convex and admits a unique solution.

- In practice, we show that the proposed approach, dubbed PowerQuant, only requires simple modifications in the quantized DNN activation functions and accumulation. Furthermore, we demonstrate through extensive experimentation that our method achieves outstanding results on various and challenging benchmarks with negligible computational overhead.

## 2 RELATED WORK

### 2.1 QUANTIZATION

In this section, we provide a background on the current state of DNNs quantization. Notice that while certain approaches are geared towards memory footprint reduction (e.g. without quantizing inputs and activations) (Chen et al., 2015; Gong et al., 2014; Han et al., 2016; Zhou et al., 2017), in what follows, we essentially focus on methods that aim at reducing the inference time. In particular, motivated by the growing concerns for privacy and security, data-free quantization methods (Banner et al., 2019; Cai et al., 2020; Choukroun et al., 2019; Fang et al., 2020; Garg et al., 2021; Zhao et al., 2019; Nagel et al., 2019; Cong et al., 2022) are emerging and have significantly improved over the recent years. The first breakthrough in data-free quantization (Nagel et al., 2019) was based on two mathematical ingenuities. First, they exploited the mathematical properties of piece-wise affine activation functions (such as e.g. ReLU based DNNs) in order to balance the per-

channel weight distributions by iteratively applying scaling factors to consecutive layers. Second, they proposed a bias correction scheme that consists in updating the bias terms of the layers with the difference between the expected quantized prediction and the original predictions. They achieved near full-precision accuracy in int8 quantization. Since this seminal work, two main categories of data-free quantization methods have emerged. First, data-generation based methods, such as Cai et al. (2020); Garg et al. (2021), that used samples generated by Generative Adversarial Networks (GANs) (Goodfellow et al., 2014) as samples to fine-tune the quantized model through knowledge distillation (Hinton et al., 2014). Nevertheless, these methods are time-consuming and require significantly more computational resources. Other methods, such as Banner et al. (2019); Choukroun et al. (2019); Fang et al. (2020); Zhao et al. (2019); Nagel et al. (2019); Cong et al. (2022), focus on improving the quantization operator but usually achieve lower accuracy. One limitation of these approaches is that they are essentially restricted to uniform quantization, while considering non-uniform mappings between the floating point and low-bit representation might be key to superior performance.

## 2.2 NON-UNIFORM QUANTIZATION

Indeed, in uniform settings, continuous variables are mapped to an equally-spaced grid in the original, floating point space. Such mapping introduces an error: however, applying such uniform mapping to an *a priori* non-uniform weight distribution is likely to be suboptimal in the general case. To circumvent this limitation, non-uniform quantization has been introduced (Banner et al., 2019; Hubara et al., 2016; Jeon et al., 2020; Wu et al., 2016; Zhang et al., 2018; Miyashita et al., 2016; Zhou et al., 2017) and (Zhang et al., 2021a; Li et al., 2019). We distinguish two categories of non-uniform quantization approaches. First, methods that introduce a code-base and require very sophisticated implementations for actual inference benefits such as Banner et al. (2019); Hubara et al. (2016); Jeon et al. (2020); Wu et al. (2016); Zhang et al. (2018). Second, methods that simply modify the quantization operator such as Miyashita et al. (2016); Zhou et al. (2017). In particular, (Zhang et al., 2021a) propose a log-quantization technique. Similarly, Li *et al.* (Li et al., 2019) use log quantization with basis 2. In both cases, in practice, such logarithmic quantization scheme changes the nature of the mathematical operations involved, with multiplications being replaced by bit shifts. Nevertheless, one limitation of this approach is that because the very nature of the mathematical operations is intrinsically altered, in practice, it is hard to leverage without dedicated hardware and implementation. Instead of transforming floating point multiplications in integer multiplications, they change floating point multiplications into bit-shifts or even look up tables (LUTs). Some of these operations are very specific to some hardware (e.g. LUTs are thought for FPGAs) and may not be well supported on most hardware. Conversely, in this work, we propose a non-uniform quantization scheme that preserves the nature of the mathematical operations by mapping floating point multiplications to standard integer multiplications. As a result, our approach boils down to simple modifications of the computations in the quantized DNN, hence allowing higher accuracies than uniform quantization methods while leading to straightforward, ready-to-use inference speed gains. Below we describe the methodology behind the proposed approach.

## 3 METHODOLOGY

Let $F$ be a trained feed forward neural network with $L$ layers, each comprising a weight tensor $W_l$. Let $Q$ be a quantization operator such that the quantized weights $Q(W_l)$ are represented on $b$ bits. The most popular such operator is the uniform one. We argue that, despite its simplicity, the choice of such a uniform operator is responsible for a significant part of the quantization error. In fact, the weights $W_l$ most often follow a bell-shaped distribution for which uniform quantization is ill-suited: intuitively, in such a case, we would want to quantize more precisely the small weights on the peak of the distribution. For this reason, the most popular non-uniform quantization scheme is logarithmic quantization, outputting superior performance. Practically speaking, however, it consists in replacing the quantized multiplications by bit-shift operations. As a result, these methods have limited adaptability as the increment speed is hardware dependent. To adress this problem, we look for the non-uniform quantization operators that preserve the nature of matrix multiplications. Formally, taking aside the rounding operation in quantization, we want to define the space $\mathcal{Q}$ of functions $Q$ such that

$$\forall Q \in \mathcal{Q}, \exists Q^{-1} \in \mathcal{Q} \quad \text{s.t.} \ \forall x, y \quad Q^{-1}(Q(x) * Q(y)) = x \times y \tag{1}$$

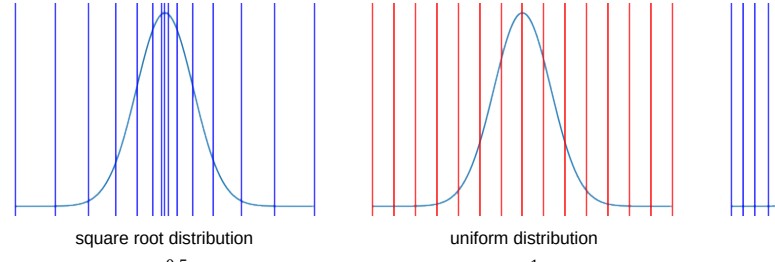

Figure 2: Influence of the power parameter $a$ on the quantized distribution for weights distributed following a Gaussian prior. In such a case, the reconstruction error is typically minimized for $a < 1$.

where $*$ is the intern composition law of the quantized space and $\times$ is the standard multiplication, and $Q, Q^{-1}$ are the quantization and de-quantization operators, respectively. In the case of uniform quantization and our work $*$ will be the multiplication while in other non-uniform works it often corresponds to other operations that are harder to leverage, e.g. bit-shift. Recall that, for now, we omit the rounding operator. The proposed PowerQuant method consists in the search for the best suited operator $Q$ for a given trained neural network and input statistics.

## 3.1 AUTOMORPHISMS OF $(R_+^*, \times)$

Let $Q$ be a transformation from $\mathbb{R}_+$ to $\mathbb{R}_+$. In this case, $*$, the intern composition law in quantized space in (1), simply denote the scalar multiplication operator $\times$ and (1) becomes $Q(x) \times Q(y) = Q(x \times y) \; \forall x, y \in \mathbb{R}_+^2$. In order to define a de-quantization operation, we need $Q^{-1}$ to be defined *i.e.* $Q$ is bijective. Thus, by definition, $Q$ is a group automorphism of $(R_+^*, \times)$. Thus, quantization operators that preserve the nature of multiplications are restricted to automorphisms of $(R_+^*, \times)$. The following lemma further restricts the search to power functions.

**Lemma 1.** *The set of continuous automorphisms of $(R_+^*, \times)$ is defined by the set of power functions* $\mathcal{Q} = \{Q : x \mapsto x^a | a \in \mathbb{R}\}$.

A proof of this result can be found in Appendix A. For the sake of clarity, we will now include the rounding operation in the quantization operators.

$$\mathcal{Q} = \left\{ Q_a : W \mapsto \left\lfloor (2^{b-1} - 1) \frac{\text{sign}(W) \times |W|^a}{\max |W|^a} \right\rceil \Big| a \in \mathbb{R} \right\} \tag{2}$$

where $W$ is a tensor and all the operations are performed element-wise. As functions of $W$, the quantization operators defined in equation 2 are (signed) power functions. Fig 2 illustrates the effect of the power parameter $a$ on quantization (vertical bars). Uniform quantization and $a = 1$ are equivalent and correspond to a quantization invariant to the weight distribution. For $a < 1$, the quantization is more fine-grained on weight values with low absolute value and coarser on high absolute values. Conversely, for $a > 1$, the quantization becomes more fine-grained on high absolute values. We now define the search protocol in the proposed search space $\mathcal{Q}$.

## 3.2 AUTOMORPHISM SEARCH AS A MINIMIZATION PROBLEM

We propose to use the error introduced by quantization on the weights as a proxy on the distance between the quantized and the original model.

**Reconstruction Error Minimization:** The operator $Q_a$ is not a bijection. Thus, quantization introduces a reconstruction error summed over all the layers of the network, and defined as follows:

$$\epsilon(F, a) = \sum_{l=1}^{L} \left\| W_l - Q_a^{-1}(Q_a(W_l)) \right\|_p \tag{3}$$

where $\| \cdot \|_p$ denotes the $L^p$ vector norm (in practice $p = 2$, see appendix B) and the de-quantization operator $Q_a^{-1}$ is defined as:

$$Q_a^{-1}(W) = \text{sign}(W) \times \left| W \times \frac{\max |W|}{2^{b-1} - 1} \right|^{\frac{1}{a}} \tag{4}$$

In practice, the problem of finding the best exponent $a^* = \text{argmin}_a \epsilon(F, a)$ in (3) is a locally convex optimization problem (Appendix C.1) which has a unique minimum (see Appendix C.2). We find the optimal value for $a$ using the Nelder–Mead method (Nelder & Mead, 1965) which solves problems for which derivatives may not be known or, in our case, are almost-surely zero (due to the rounding operation). In practice, more recent solvers are not required in order to reach the optimal solution (see Appendix D). Lastly, we discuss the limitations of the proposed metric in Appendix H.

## 3.3 Fused De-Quantization and Activation Function

Based on equation 2, the quantization process of the weights necessitates the storage and multiplication of $W$ along with a signs tensor, which is memory and computationally intensive. For the weights, however, this can be computed once during the quantization process, inducing no overhead during inference. As for activations, we do not have to store the sign of ReLU activations as they are always positive. In this case, the power function has to be computed at inference time (see algorithm 2). However, it can be efficiently computed Kim et al. (2021), using Newton's method to approximate continuous functions in integer-only arithmetic. This method is very efficient in practice as it converges in 2 steps for low bit representations (four steps for int32). Thus, PowerQuant leads to significant accuracy gains with limited computational overhead. Conversely, for non-ReLU feed forward networks such as EfficientNets (SiLU) or Image Transformers (GeLU), activations are signed. This can be tackled using asymmetric quantization which consists in the use of a zero-point. In general, asymmetric quantization allows one to have a better coverage of the quantized values support. In our case, we use asymmetric quantization to work with positive values only. Formally, for both SiLU and GeLU, the activations are analytically bounded below by $C_{\text{SiLU}} = 0.27846$ and $C_{\text{GeLU}} = 0.169971$ respectively. Consequently, assuming a layer with SiLU activation with input $x$ and weights $W$, we have:

$$Q_a^{-1}\left(Q_a(x + C_{\text{SiLU}})Q_a(W)\right) \approx \left((x + C_{\text{SiLU}})^a W^a\right)^{\frac{1}{a}} = xW + C_{\text{SiLU}}W \tag{5}$$

The bias term $C_{\text{SiLU}}W$ induces a very slight computation overhead which is standard in asymmetric quantization. We provide a detailed empirical evaluation of this cost in Appendix G. Using the adequate value for the bias corrector, we can generalize equation 5 to any activation function $\sigma$. The quantization process and inference with the quantized DNN are summarized in Algorithm 1 and 2. The proposed representation is fully compatible with integer multiplication as defined in Jacob et al. (2018), thus it is fully compatible with integer only inference (see appendix F for more details).

---

**Algorithm 1** Weight Quantization Algorithm

---

**Require:** trained neural network $F$ with $L$ layers to quantize, number of bits $b$
$\quad a \leftarrow \text{solver}(\min\{\text{error}(F, a)\})$             $\triangleright$ in practice we use the Nelder–Mead method
$\quad$ **for** $l \in \{1, \dots, L\}$ **do**
$\quad\quad W_{\text{sign}} \leftarrow \text{sign}(W_l)$           $\triangleright$ save the sign of the scalar values in $W$
$\quad\quad W_l \leftarrow W_{\text{sign}} \times |W_l|^a$           $\triangleright$ power transformation
$\quad\quad s \leftarrow \frac{\max |W_l|}{2^{b-1}-1}$           $\triangleright$ get quantization scale
$\quad\quad Q : W_l \mapsto \left\lfloor \frac{W_l}{s} \right\rceil$ and $Q^{-1} : W \mapsto W_{\text{sign}} \times |W \times s|^{\frac{1}{a}}$      $\triangleright$ qdefine $Q$ and $Q^{-1}$
$\quad$ **end for**

---

## 4 Experiments

In this section, we empirically validate our method. First, we discuss the optimization of the exponent parameter $a$ of PowerQuant using the reconstruction error, showing its interest as a proxy for the quantized model accuracy from an experimental standpoint. We show that the proposed approach preserves this reconstruction error significantly better, allowing a closer fit to the original weight distribution through non-uniform quantization. Second, we show through a variety of benchmarks that the proposed approach significantly outperforms state-of-the-art data-free methods, thanks to more efficient power function quantization with optimized exponent. Third, we show that the proposed approach comes at a negligible cost in term of inference speed.

---

**Algorithm 2** Simulated Inference Algorithm

---

**Require:** trained neural network $F$ quantized with $L$ layers, input $X$ and exponent $a^*$

  **for** $l \in \{1, \dots, L\}$ **do**

      $X \leftarrow X^{a^*}$                                     $\triangleright$ $X$ is assumed positive (see equation (5))

      $X^Q \leftarrow \lfloor X s_X \rceil$                               $\triangleright$ where $s_X$ is a scale in the input range

      $O \leftarrow F_l(X^Q)$                           $\triangleright$ where we applied $\cdot\frac{1}{a^*}$ at the accumulation

      $X \leftarrow \left(\frac{\sigma(O)}{s_X s_W}\right)$                   $\triangleright$ where $\sigma$ is the activation function and $s_W$ the weight scale

  **end for**

---

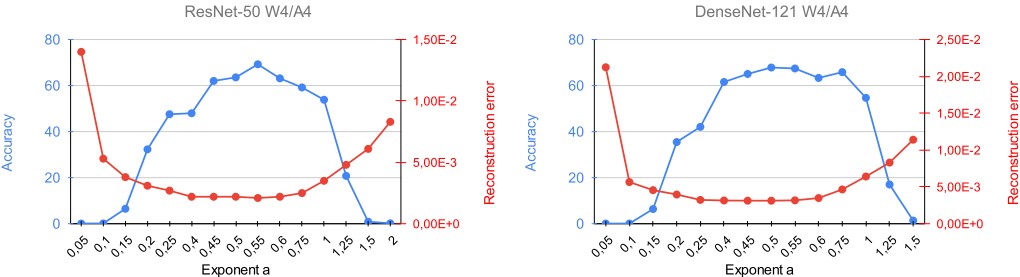

Figure 3: Accuracy/reconstruction error relationship for ResNet and DenseNet quantized in W4/A4.

## 4.1 DATASETS AND IMPLEMENTATION DETAILS

We validate the proposed PowerQuant method on ImageNet classification (Deng et al., 2009) ($\approx$ 1.2M images train/50k test). In our experiments we used pre-trained MobileNets (Sandler et al., 2018), ResNets (He et al., 2016), EfficientNets (Tan & Le, 2019) and DenseNets (Huang et al., 2017). We used Tensorflow implementations of the baseline models from official repositories, achieving standard baseline accuracies. The quantization process was done using Numpy library. Activations are quantized as unsigned integers and weights are quantized using a symmetric representation. We fold batch-normalization layers as in Yvinec et al. (2022a). We performed ablation study using the uniform quantization operator over weight values from Krishnamoorthi (2018) and logarithmic quantization from Miyashita et al. (2016). For our comparison with state-of-the-art approaches in data-free quantization, we implemented the more complex quantization operator from SQuant (Cong et al., 2022). To compare with strong baselines, we also implement bias correction (Nagel et al., 2019) (which measures the expected difference between the outputs of the original and quantized models and updates the biases terms to compensate for this difference) as well as input weight quantization (Nagel et al., 2019).

## 4.2 EXPONENT PARAMETER FITTING

Fig 3 illustrates the evolution of both the accuracy of the whole DNN and the reconstruction error summed over all the layers of the network, as functions of the exponent parameter $a$. Our target is the highest accuracy with respect to the value of $a$: however, in a data-free context, we only have access to the reconstruction error. Nevertheless, as shown on Fig 3, these metrics are strongly anti-correlated. Furthermore, while the reconstruction curve is not convex it behaves well for simplex based optimization method such as the Nelder-Mead method. This is due to two properties: locally convex (Appendix C.1) and has a unique minimum (Appendix C.2).

Empirically, optimal values $a^*$ for the exponent parameter are centered on 0.55, which approximately corresponds to the first distribution in Fig 2. Still, as shown on Table 1 we observe some variations on the best value for $a$ which motivates the optimization of $a$ for each network and bit-width. Furthermore, our results provide a novel insight on the difference between pruning and quantization. In the pruning literature (Han et al., 2015; Frankle & Carbin, 2018; Molchanov et al., 2019), the baseline method consists in setting the smallest scalar weight values to zero and keeping unchanged the highest non-zero values, assuming that small weights contribute less to the network prediction. In a similar vein, logarithmic or power quantization with $a > 1$ roughly quantizes

Table 1: Comparison between logarithmic, uniform and the proposed quantization scheme on ResNet 50 trained for ImageNet classification task. We report for different quantization configuration (weights noted W and activations noted A) both the top1 accuracy and the reconstruction error (equation 3).

| Architecture | Method | W-bit | A-bit | $a^*$ | Accuracy | Reconstruction Error |
|---|---|---|---|---|---|---|
| | Baseline | 32 | 32 | - | 76.15 | - |
| | uniform | 8 | 8 | 1 | 76.15 | $1.1 \times 10^{-4}$ |
| | logarithmic | 8 | 8 | - | 76.12 | $2.0 \times 10^{-4}$ |
| ResNet 50 | PowerQuant | 8 | 8 | 0.55 | **76.15** | $\mathbf{1.0} \times 10^{-4}$ |
| | uniform | 4 | 4 | 1 | 54.68 | $3.5 \times 10^{-3}$ |
| | logarithmic | 4 | 4 | - | 57.07 | $2.1 \times 10^{-3}$ |
| | PowerQuant | 4 | 4 | 0.55 | **70.29** | $\mathbf{1.9} \times 10^{-3}$ |

Table 2: Comparison between state-of-the-art post training quantization techniques on ResNet 50 on ImageNet. We distinguish methods relying on data (synthetic or real) or not. In addition to being fully data-free, our approach significantly outperforms existing methods.

| Architecture | Method | Data | W-bit | A-bit | Accuracy | gap |
|---|---|---|---|---|---|---|
| | Baseline | - | 32 | 32 | 76.15 | - |
| | DFQ Nagel et al. (2019) | No | 8 | 8 | 75.45 | -0.70 |
| | ZeroQ Cai et al. (2020) | Synthetic | 8 | 8 | 75.89 | -0.26 |
| | DSG Zhang et al. (2021b) | Synthetic | 8 | 8 | 75.87 | -0.28 |
| | GDFQ Xu et al. (2020) | Synthetic | 8 | 8 | 75.71 | -0.44 |
| | SQuant Cong et al. (2022) | No | 8 | 8 | 76.04 | -0.11 |
| ResNet 50 | PowerQuant | No | 8 | 8 | **76.15** | **0.00** |
| | DFQ Nagel et al. (2019) | No | 4 | 4 | 0.10 | -76.05 |
| | ZeroQ Cai et al. (2020) | Synthetic | 4 | 4 | 7.75 | -68.40 |
| | DSG Zhang et al. (2021b) | Synthetic | 4 | 4 | 23.10 | -53.05 |
| | GDFQ Xu et al. (2020) | Synthetic | 4 | 4 | 55.65 | -20.50 |
| | SQuant Cong et al. (2022) | No | 4 | 4 | 68.60 | -7.55 |
| | PowerQuant | No | 4 | 4 | **70.53** | **-5.62** |

(almost zeroing it out) small scalar values to better preserve the precision on larger values. In practice, in our case, lower reconstruction errors, and better accuracies, are achieved by setting $a < 1$: this suggests that the assumption behind pruning can't be straightforwardly applied to quantization, where in fact we argue that finely quantizing smaller weights is paramount to preserve the patterns learned at each layer, and the representation power of the whole network.

Another approach that puts more emphasis on the nuances between low valued weights is logarithmic based non-uniform quantization. In Table 1 and Appendix E, we compare the proposed power method to both uniform and logarithmic approaches. By definition, the proposed power method necessarily outperforms the uniform method in every scenario as uniform quantization is included in the search space. For instance, in int4, the proposed method improves the accuracy by 13.22 points on ResNet 50. This improvement can also be attributed to a better input quantization of each layer, especially on ResNet 50 where the gap in the reconstruction error (over the weights) is smaller.

## 4.3 COMPARISON WITH DATA-FREE QUANTIZATION METHODS

In table 2, we report the performance of several data-free quantization approaches on ResNet 50. Although no real training data is involved in these methods, some approaches such as ZeroQ (Cai et al., 2020), DSG (Zhang et al., 2021b) or GDFQ (Xu et al., 2020) rely on data generation (DG) in order to calibrate parameters of the method or to apply fine-tuning to preserve the accuracy through quantization. As shown in table 2, in the W8/A8 setup, the proposed PowerQuant method outperforms other data-free solutions, fully preserving the accuracy of the floating point model. The gap is even wider on the more challenging low bit quantization W4/A4 setup, where the PowerQuant improves the accuracy by 1.93 points over SQuant (Cong et al., 2022) and by 14.88 points over GDFQ. This shows the effectiveness of the method on ResNet 50. We provide more results on DenseNet (Huang et al., 2017), MobileNet (Sandler et al., 2018), Efficient Net (Tan & Le, 2019) in Appendix

Table 3: Comparison of data-free quantization methods on ViT and DeiT trained on ImageNet.

| model | method | W / A | accuracy |
|---|---|---|---|
| ViT | baseline | -/- | 78.05% |
| | DFQ (ICCV 2019) | 8/8 | 70.33% |
| | SQuant (ICLR 2022) | 8/8 | 68.85% |
| | PSAQ (arxiv 2022) | 8/8 | 37.36% |
| | PowerQuant | 8/8 | **77.46%** |
| | DFQ (ICCV 2019) | 4/8 | 66.63% |
| | SQuant (ICLR 2022) | 4/8 | 64.62% |
| | PSAQ (arxiv 2022) | 4/8 | 25.34% |
| | PowerQuant | 4/8 | **75.24%** |

(a) Evaluation for ViT Base

| model | method | W / A | accuracy |
|---|---|---|---|
| DeiT T | baseline | -/- | 72.21% |
| | DFQ (ICCV 2019) | 8/8 | 71.32% |
| | SQuant (ICLR 2022) | 8/8 | 71.11% |
| | PSAQ (arxiv 2022) | 8/8 | 71.56% |
| | PowerQuant | 8/8 | **72.23%** |
| | DFQ (ICCV 2019) | 4/8 | 67.71% |
| | SQuant (ICLR 2022) | 4/8 | 67.58% |
| | PSAQ (arxiv 2022) | 4/8 | 65.57% |
| | PowerQuant | 4/8 | **69.77%** |

(b) Evaluation for DeiT Tiny

| model | method | W / A | accuracy |
|---|---|---|---|
| DeiT S | baseline | -/- | 79.85% |
| | DFQ (ICCV 2019) | 8/8 | 78.76% |
| | SQuant (ICLR 2022) | 8/8 | 78.94% |
| | PSAQ (arxiv 2022) | 8/8 | 76.92% |
| | PowerQuant | 8/8 | **79.33%** |
| | DFQ (ICCV 2019) | 4/8 | 76.75% |
| | SQuant (ICLR 2022) | 4/8 | 76.61% |
| | PSAQ (arxiv 2022) | 4/8 | 73.23% |
| | PowerQuant | 4/8 | **78.16%** |

(c) Evaluation for DeiT Small

| model | method | W / A | accuracy |
|---|---|---|---|
| DeiT B | baseline | -/- | 81.85% |
| | DFQ (ICCV 2019) | 8/8 | 80.72% |
| | SQuant (ICLR 2022) | 8/8 | 80.60% |
| | PSAQ (arxiv 2022) | 8/8 | 79.10% |
| | PowerQuant | 8/8 | **81.26%** |
| | DFQ (ICCV 2019) | 4/8 | 79.41% |
| | SQuant (ICLR 2022) | 4/8 | 79.21% |
| | PSAQ (arxiv 2022) | 4/8 | 77.05% |
| | PowerQuant | 4/8 | **80.67%** |

(d) Evaluation for DeiT Base

J. These results demonstrate the versatility of the method on both large and very compact convnets. In summary, the proposed PowerQuant vastly outperforms other data-free quantization schemes. Last but not least, when compared to recent QAT methods such as OCTAV Sakr et al. (2022), PowerQuant achieves competitive results on both ResNets and MobileNets using either both static or dynamic quantization. This is remarkable since PowerQuant does not involve any fine-tuning of the network. We provide more details on this benchamrk in Appendix I. In what follows, we evaluate PowerQuant on recent transformer architectures for both image and language applications.

## 4.4 EVALUATION ON TRANSFORMER ARCHITECTURES

In Table 3, we quantized the weight tensors of a ViT Dosovitskiy et al. (2021) with 85M parameters and baseline accuracy $\approx 78$ as well as DeiT T,S and B Touvron et al. (2021) with baseline accuracies 72.2, 79.9 and 81.8 and $\approx 5M$, $\approx 22M$, $\approx 87M$ parameters respectively. Similarly to ConvNets, the image transformer is better quantized using PowerQuant rather than standard uniform quantization schemes such as DFQ. Furthermore, more complex and recent data-free quantization schemes such as SQuant, tend to under-perform on the novel Transformer architectures as compared to ConvNets. This is not the case for PowerQuant which maintains its very high performance even in low bit representations. This is best illustrated on ViT where PowerQuant W4/A8 out performs both DFQ and SQuant even when they are allowed 8 bits for the weights (W8/A8) by a whopping 4.91 points. The proposed PowerQuant even outperforms methods dedicated to transformer quantization such as PSAQ Li et al. (2022) on every image transformer tested. We further compare the proposed power quantization, in W4/A8, on natural language processing (NLP) tasks and report results in Table 4. We evaluate a BERT model (Devlin et al., 2018) on GLUE (Wang et al., 2018) and report both the original (reference) and our reproduced (baseline) results. We compare the three quantization processes: uniform, logarithmic and PowerQuant. Similarly to computer vision tasks, the power quantization outperforms the other methods in every instances which further confirms its ability to generalize well to transformers and NLP tasks. In what follows, we show experimentally that our approach induces very negligible overhead at inference time, making this accuracy enhancement virtually free from a computational standpoint.

Table 4: Complementary Benchmarks on the GLUE task with the BERT transformer architecture quantized in W4/A8. We provide the original performance (from the article) as well as our reproduced results (baseline).

| task | original | baseline | uniform | log | SQuant | PowerQuant |
|------|----------|----------|---------|-----|--------|-----------|
| CoLA | 49.23 | 47.90 | 45.60 | 45.67 | 46.88 | **47.11** |
| SST-2 | 91.97 | 92.32 | 91.81 | 91.53 | 91.09 | **92.23** |
| MRPC | 89.47/85.29 | 89.32/85.41 | 88.24/84.49 | 86.54/82.69 | 88.78/85.24 | **89.26/85.34** |
| STS-B | 83.95/83.70 | 84.01/83.87 | 83.89/83.85 | **84.01**/83.81 | 83.80/83.65 | **84.01/83.87** |
| QQP | 88.40/84.31 | 90.77/84.65 | 89.56/83.65 | 90.30/84.04 | 90.34/84.32 | **90.61/84.45** |
| MNLI | 80.61/81.08 | 80.54/80.71 | 78.96/79.13 | 78.96/79.71 | 78.35/79.56 | **79.02/80.28** |
| QNLI | 87.46 | 91.47 | 89.36 | 89.52 | 90.08 | **90.23** |
| RTE | 61.73 | 61.82 | 60.96 | 60.46 | 60.21 | **61.45** |
| WNLI | 45.07 | 43.76 | 39.06 | 42.19 | 42.56 | **42.72** |

Table 5: ACE cost of the overhead computations introduced by PowerQuant.

| Architecture | overhead cost | accuracy in W6/A6 |
|--------------|---------------|-------------------|
| ResNet 50 | 0.63% | 75.07 |
| DenseNet 121 | 0.97% | 72.71 |
| MobileNet V2 | 0.57% | 52.20 |
| EfficientNet B0 | 0.80% | 58.24 |

## 4.5 INFERENCE COST AND PROCESSING TIME

The ACE metrics was recently introduced in Zhang et al. (2022) to provide a hardware-agnostic measurement of the overhead computation cost in quantized neural networks. In Table 5, we evaluate the cost in the inference graph due to the change in the activation function. We observe very similar results to Table 17. The proposed changes are negligible in terms of computational cost on all tested networks. Furthermore, DenseNet has the highest cost due to its very dense connectivity. On the other hand, using this metric it seems that the overhead cost due to the zero-point technique from section 3.3 for EfficientNet has no significant impact as compared to MobileNet and ResNet. In addition, we discuss the inference and processing cost of PowerQuant on specific hardware using dedicated tools in Appendix K.

## 5 CONCLUSION

In this paper, we pinpointed the uniformity of the quantization as a limitation of existing data-free methods. To address this limitation, we proposed a novel data-free method for non-uniform quantization of trained neural networks for computer vision tasks, with an emphasis on not changing the nature of the mathematical operations involved (e.g. matrix multiplication). This led us to search among the continuous automorphisms of $(\mathbb{R}_+^*, \times)$, which are restricted to the power functions $x \to x^a$. We proposed an optimization of this exponent parameter based upon the reconstruction error between the original floating point weights and the quantized ones. We show that this procedure is locally convex and admits a unique solution. At inference time, the proposed approach, dubbed PowerQuant, involves only very simple modifications in the quantized DNN activation functions. We empirically demonstrate that PowerQuant allows a closer fit to the original weight distributions compared with uniform or logarithmic baselines, and significantly outperforms existing methods in a variety of benchmarks with only negligible computational overhead at inference time. In addition, we also discussed and addressed some of the limitations in terms of optimization (per-layer or global) and generalization (non-ReLU networks). Future work involves the search of a better proxy error as compared with the proposed weight reconstruction error as well as the extension of the search space to other composition laws of $R_+$ that are suited for efficient calculus and inference.

### ACKNOWLEDGMENTS

This work has been supported by the french National Association for Research and Technology (ANRT), the company Datakalab (CIFRE convention C20/1396) and by the French National Agency (ANR) (FacIL, project ANR-17-CE33-0002). This work was granted access to the HPC resources of IDRIS under the allocation 2022-AD011013384 made by GENCI.

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

## A    PROOF OF LEMMA 1

In this section, we provide a simple proof for lemma 1 as well as a discussion on the continuity hypothesis.

*Proof.* We have that $\forall x \in \mathbb{R}_+, Q(x) \times Q(0) = Q(0)$ and $\forall x \in \mathbb{R}_+, Q(x) \times Q(1) = Q(x)$ which induces that $Q$ is either the constant 1 or $Q(0) = 0$ and $Q(1) = 1$. Because $Q$ is an automorphism we can eliminate the first option. Now, we will demonstrate that $Q$ is necessarily a power function. Let $n$ be an integer, then

$$Q(x^n) = Q(x) \times Q(x^{n-1}) = Q(x)^2 \times Q(x^{n-2}) = \cdots = Q(x)^n. \tag{6}$$

Similarly, for fractions, we get $Q(x^{\frac{1}{n}}) \times \cdots \times Q(x^{\frac{1}{n}}) = Q(x) \Leftrightarrow Q(x^{\frac{1}{n}}) = Q(x)^{\frac{1}{n}}$. Assuming $Q$ is continuous, we deduce that for any rational $a \in \mathbb{R}$, we have

$$Q(x^a) = Q(x)^a \tag{7}$$

In order to verify that the solution is limited to power functions, we use a *reductio ad absurdum*. Assume $Q$ is not a power function. Therefore, there exists $(x, y) \in \mathbb{R}_+^2$ and $a \in \mathbb{R}$ such that $Q(x) \neq x^a$ and $Q(y) = y^a$. By definition of the logarithm, there exists $b$ such that $x^b = y$. We get the following contradiction, from (7),

$$\begin{cases} Q(x^{b^a}) = Q(y^a) = y^a \\ Q(x^{b^a}) = Q(x^{ab}) = Q(x^a)^b \neq (x^{ab} = y^a) \end{cases} \tag{8}$$

Consequently, the suited functions $Q$ are limited to power functions *i.e.* $\mathcal{Q} = \{Q : x \mapsto x^a | a \in \mathbb{R}\}$. $\qquad\square$

We would also like to put the emphasis on the fact that there are other Automorphisms of $(\mathbb{R}, \times)$. However, the construction of such automorphisms require the axiom of choice Herrlich (2006). Such automorphisms are not applicable in our case which is why the key constraint is being an automorphism rather than the continuous property.

## B    NORM SELECTION

In the minimization objective, we need to select a norm to apply. In this section, we provide theoretical arguments in favor of the $l^2$ vector norm. Let $F$ be a feed forward neural network with $L$ layers to quantize, each defined by a set of weights $W_l = (w_l)_{i,j} \in \mathbb{R}^{n_l \times m_l}$ and bias $b_l \in \mathbb{R}^{n_l}$. We note $(\lambda_l^{(i)})_i$ the eigenvalues associated with $W_l$. We want to study the distance $d(F, F_a)$ between the predictive function $F$ and its quantized version $F_a$ defined as

$$d(F, F_a) = \max_{x \in \mathcal{D}} \|F(x) - F_a(x)\|_p \tag{9}$$

where $\mathcal{D}$ is the domain of $F$. We prove that minimizing the reconstruction error with respect to $a$ is equivalent to minimizing $d(F, F_a)$ with respect to $a$. Assume $L = 1$ for the sake of simplicity and we drop the notation $l$. With the proposed PowerQuant method, we minimize the vector norm

$$\|W - Q_a^{-1}(Q_a(W))\|_p^p = \sum_{i <= n} \max_{j <= m} |w_{i,j} - Q_a^{-1}(Q_a(w_{i,j}))|^p \tag{10}$$

For $p = 2$, the euclidean norm is equal to the spectral norm, thus minimizing $\|W - Q_a^{-1}(Q_a(W))\|_2$ is equivalent to minimizing $d(F, F_a)$ for $L = 1$. However, we know that minimizing for another value of $p$ may result in a different optimal solution and therefore not necessarily minimize $d(F, F_a)$.

In the context of data-free quantization, we want to avoid uncontrollable changes on $F$, which is why we recommend the use of $p = 2$.

## C  MATHEMATICAL PROPERTIES

### C.1  LOCAL CONVEXITY

We prove that the minimization problem defined in equation 3 is locally convex around the solution $a^*$. Formally we prove that

$$x \mapsto \left\| x - Q_a^{-1}\left(Q_a(x)\right) \right\|_p \tag{11}$$

is locally convex around $a^*$ defined as $\arg \min_a \left\| x - Q_a^{-1}\left(Q_a(x)\right) \right\|_p$.

**Lemma 2.** *The minimization problem defined as*

$$\arg \min_a \left\{ \left\| x - Q_a^{-1}\left(Q_a(x)\right) \right\|_p \right\} \tag{12}$$

*is locally convex around any solution $a^*$.*

*Proof.* We recall that $\frac{\partial x^a}{\partial a} = x^a \log(x)$. The function $\left\| x - Q_a^{-1}\left(Q_a(x)\right) \right\|$ is differentiable. We assume $x \in \mathbb{R}$, then we can simplify the sign functions (assume $x$ positive without loss of generality) and note $y = \max |x|$, then

$$\frac{\partial Q_a^{-1}\left(Q_a(x)\right)}{\partial a} = \frac{\partial \left| \left\lfloor (2^{b-1} - 1)\frac{x^a}{y^a} \right\rfloor \frac{y^a}{2^{b-1}-1} \right|^{\frac{1}{a}}}{\partial a}. \tag{13}$$

This simplifies to

$$\frac{\partial Q_a^{-1}\left(Q_a(x)\right)}{\partial a} = y\frac{\partial \left( \frac{\left\lfloor B\left(\frac{x}{y}\right)^a \right\rfloor}{B} \right)^{\frac{1}{a}}}{\partial a}, \tag{14}$$

with $B = 2^{b-1} - 1$. By using the standard differentiation rules, we know that the rounding operator has a zero derivative a.e.. Consequently we get,

$$\frac{\partial Q_a^{-1}\left(Q_a(x)\right)}{\partial a} = -a^2 y \left( \frac{\left\lfloor B\left(\frac{x}{y}\right)^a \right\rfloor}{B} \right)^{\frac{1}{a}} \log \left( \frac{\left\lfloor B\left(\frac{x}{y}\right)^a \right\rfloor}{B} \right). \tag{15}$$

Now we can compute the second derivative of $Q_a^{-1}\left(Q_a(x)\right)$,

$$\frac{\partial^2 Q_a^{-1}\left(Q_a(x)\right)}{\partial a^2} = a^4 y \left( \frac{\left\lfloor B\left(\frac{x}{y}\right)^a \right\rfloor}{B} \right)^{\frac{1}{a}} \log^2 \left( \frac{\left\lfloor B\left(\frac{x}{y}\right)^a \right\rfloor}{B} \right). \tag{16}$$

From this expression, we derive the second derivative, using the property $(f \circ g)'' = f'' \circ g \times g'^2 + f' \circ g \times g''$ and the derivatives $|\cdot|^{\frac{1}{p}'} = \frac{x|x|^{\frac{1}{p}-2}}{p}$ and $|\cdot|^{\frac{1}{p}''} = \frac{1-p}{p^2}\frac{|x|^{\frac{1}{p}}}{x^2}$, then for any $x_i \in x$

$$
\begin{aligned}
\frac{\partial^2 \left| x_i - Q_a^{-1}\left(Q_a(x_i)\right) \right|}{\partial a^2} =& \frac{1-p}{p^2} \frac{|x_i - Q_a^{-1}(Q_a(x_i)|^{\frac{1}{p}}}{(x_i - Q_a^{-1}(Q_a(x_i))^2} \left( \frac{\partial Q_a^{-1}\left(Q_a(x)\right)}{\partial a} \right)^2 \\
&+ \frac{(x_i - Q_a^{-1}(Q_a(x_i))|x_i - Q_a^{-1}(Q_a(x_i)|^{\frac{1}{p}-2}}{p} \frac{\partial^2 Q_a^{-1}\left(Q_a(x)\right)}{\partial a^2}
\end{aligned}
\tag{17}
$$

We now note the first term in the previous addition $T_1 = \frac{1-p}{p^2} \frac{|x_i-Q_a^{-1}(Q_a(x_i)|^{\frac{1}{p}}}{(x_i-Q_a^{-1}(Q_a(x_i))^2} \left( \frac{\partial Q_a^{-1}(Q_a(x))}{\partial a} \right)^2$ and the second term as a product of $T_2 = \frac{(x_i-Q_a^{-1}(Q_a(x_i))|x_i-Q_a^{-1}(Q_a(x_i)|^{\frac{1}{p}-2}}{p}$ times $T_3 = \frac{\partial^2 Q_a^{-1}(Q_a(x))}{\partial a^2}$. We know that $T_1 > 0$ and $T_3 > 0$, consequently, and $T_2$ is continuous in $a$. At $a^*$ the terms with $|x_i - Q_a^{-1}\left(Q_a(x_i)\right)|$ are negligible in comparison with $\frac{\partial^2 Q_a^{-1}(Q_a(x))}{\partial a^2}$ and $\left( \frac{\partial Q_a^{-1}(Q_a(x))}{\partial a} \right)^2$.

Consequently, there exists an open set around $a^*$ where $T_1 > |T_2|T_3$, and $\frac{\partial^2 \left| x_i - Q_a^{-1}(Q_a(x_i)) \right|}{\partial a^2} > 0$. This concludes the proof. $\qquad \square$

Table 6: Minimization of the reconstruction error on a MobileNet V2 for W6/A6 quantization with different solvers.

| Solver | $a^*$ | reconstruction error | accuracy |
|---|---|---|---|
| Nelder-Mead | 0.750 | 1.12 | 64.248 |
| Powell (Powell, 1964) | 0.744 | 1.10 | 64.104 |
| COBYLA (Conn et al., 1997) | 0.752 | 1.11 | 64.364 |

## C.2 UNIQUENESS OF THE SOLUTION

In this section we provide the elements of proof on the uniqueness of the solution of the minimization of the quantization reconstruction error.

**Lemma 3.** *The minimization problem over $x \in \mathbb{R}^N$ defined as*

$$\arg \min_a \left\{ \left\| x - Q_a^{-1} \left( Q_a(x) \right) \right\|_p \right\} \tag{18}$$

*has almost surely a unique global minimum $a^*$.*

*Proof.* We assume that $x$ can not be exactly quantized, *i.e.* $\min_a \left\{ \left\| x - Q_a^{-1} \left( Q_a(x) \right) \right\|_p \right\} > 0$ which is true almost everywhere. We use a *reductio ad absurdum* and assume that there exist two optimal solutions $a_1$ and $a_2$ to the optimization problem. We expand the expression $\left\| x - Q_a^{-1} \left( Q_a(x) \right) \right\|_p$ and get

$$\left\| x - Q_a^{-1} \left( Q_a(x) \right) \right\|_p = \left\| x - \left\lfloor \left( 2^{b-1} - 1 \right) \frac{\text{sign}(x) \times |x|^a}{\max |x|^a} \right\rceil \frac{\max |x|^a}{2^{b-1} - 1} \right\|^{\frac{1}{a}} \text{sign}(x) \right\|_{\cdot p} \tag{19}$$

We note the rounding term $R_a$ and get

$$\left\| x - Q_a^{-1} \left( Q_a(x) \right) \right\|_p = \left\| x - \left| R_a \frac{\max |x|^a}{2^{b-1} - 1} \right|^{\frac{1}{a}} \text{sign}(x) \right\|_p . \tag{20}$$

Assume $R_{a_1} = R_{a_2} = R$, the minimization problem $\arg \min_a \left\| x - \left| R \frac{\max |x|^a}{2^{b-1} - 1} \right|^{\frac{1}{a}} \text{sign}(x) \right\|_p$ is convex and has a unique solution, thus $a_1 = a_2$. Now assume $R_{a_1} \neq R_{a_2}$.

Let's denote $D(R)$ the domain of power values $a$ over which we have $\left\lfloor \left( 2^{b-1} - 1 \right) \frac{\text{sign}(x) \times |x|^a}{\max |x|^a} \right\rceil = R$. If there is a value $a$ outside of $D(R_{a_1}) \cup D(R_{a_2})$ such that $R'$ has each of its coordinate strictly between the coordinates of $R_{a_1}$ and $R_{a_2}$, then, without loss of generality, assume that at least half of the coordinates of $R_{a_1}$ are further away from the corresponding coordinates of $x$ than one quantization step. This implies that there exists a value $a'$ in $D(R')$ such that $\left\| x - Q_{a'}^{-1} \left( Q_{a'}(x) \right) \right\|_p < \left\| x - Q_{a_1}^{-1} \left( Q_{a_1}(x) \right) \right\|_p$. which goes against our hypothesis. Thus, there are up to $N$ possible values for $R$ that minimize the problem which happens iff $x$ satisfies at least one coordinate can be either ceiled or floored by the rounding. The set defined by this condition has a zero measure. □

## D SOLVER FOR MINIMIZATION

In the main article we state that we can use Nelder-Mead (Nelder & Mead, 1965) solver to find the optimal $a^*$. We tested several other solvers and report the results in Table 6. The empirical results show that basically any popular solver can be used, and that the Nelder-Mead solver is sufficient for the minimization problem.

## E COMPARISON BETWEEN LOG, NAIVE AND POWER QUANTIZATION COMPLEMENTARY RESULTS

To complement the results provided in the main paper on ResNet 50, we list in Table 7 more quantization setups on ResNet 50 as well as DenseNet 121. To put it in a nutshell, The proposed power

Table 7: Comparison between logarithmic, uniform and the proposed quantization scheme on ResNet 50 and DenseNet 121 trained for ImageNet classification task. We report for different quantization configuration (weights noted W and activations noted A) both the top1 accuracy and the reconstruction error (equation 3).

| Architecture | Method | W-bit | A-bit | $a^*$ | Accuracy | Reconstruction Error |
|---|---|---|---|---|---|---|
| ResNet 50 | Baseline | 32 | 32 | - | 76.15 | - |
| | uniform | 8 | 8 | 1 | 76.15 | $1.1 \times 10^{-4}$ |
| | logarithmic | 8 | 8 | - | 76.12 | $2.0 \times 10^{-4}$ |
| | PowerQuant | 8 | 8 | 0.55 | **76.15** | $\mathbf{1.0} \times 10^{-4}$ |
| | uniform | 6 | 6 | 1 | 75.07 | $8.0 \times 10^{-4}$ |
| | logarithmic | 6 | 6 | - | 75.37 | $4.6 \times 10^{-4}$ |
| | power (ours) | 6 | 6 | 0.50 | **75.95** | $\mathbf{4.3} \times 10^{-4}$ |
| | uniform | 4 | 4 | 1 | 54.68 | $3.5 \times 10^{-3}$ |
| | logarithmic | 4 | 4 | - | 57.07 | $2.1 \times 10^{-3}$ |
| | PowerQuant | 4 | 4 | 0.55 | **70.29** | $\mathbf{1.9} \times 10^{-3}$ |
| DenseNet 121 | Baseline | 32 | 32 | - | 75.00 | - |
| | uniform | 8 | 8 | 1 | 75.00 | $2.8 \times 10^{-4}$ |
| | logarithmic | 8 | 8 | - | 74.91 | $2.5 \times 10^{-4}$ |
| | PowerQuant | 8 | 8 | 0.60 | **75.00** | $\mathbf{2.2} \times 10^{-4}$ |
| | uniform | 6 | 6 | 1 | 74.47 | $1.1 \times 10^{-3}$ |
| | logarithmic | 6 | 6 | - | 72.71 | $1.0 \times 10^{-3}$ |
| | power (ours) | 6 | 6 | 0.50 | **74.84** | $\mathbf{0.7} \times 10^{-3}$ |
| | uniform | 4 | 4 | 1 | 54.83 | $4.7 \times 10^{-3}$ |
| | logarithmic | 4 | 4 | - | 5.28 | $4.8 \times 10^{-3}$ |
| | PowerQuant | 4 | 4 | 0.55 | **68.04** | $\mathbf{3.1} \times 10^{-3}$ |

quantization systematically achieves significantly higher accuracy and lower reconstruction error than the logarithmic and uniform quantization schemes. On a side note, the poor performance of the logarithmic approach on DenseNet 121 can be attributed to the skewness of the weight distributions. Formally, ResNet 50 and DenseNet 121 weight values show similar average standard deviations across layers ($0.0246$ and $0.0264$ respectively) as well as similar kurtosis ($6.905$ and $6.870$ respectively). However their skewness are significantly different: $0.238$ for ResNet 50 and more than twice as much for DenseNet 121, with $0.489$. The logarithmic quantization, that focuses on very small value is very sensible to asymmetry which explains the poor performance on DenseNet 121. In contrast, the proposed method offers a robust performance in all situations.

## F    How to perform Matrix Multiplication with PowerQuant

The proposed PowerQuant method preserves the multiplication operations, *i.e.* a multiplication in the floating point space remains a multiplication in the quantized space (integers). This allows one to leverage current implementations of uniform quantization available on most hardware Gholami et al. (2021); Zhou et al. (2016). However, while PowerQuant preserves multiplications it doesn't preserve additions which are significantly less costly than multiplications. Consequently, in order to infer under the PowerQuant transformation, instead of accumulating the quantized products, as done in standard quantization Jacob et al. (2018), one need to accumulate the powers of said products. Formally, let's consider two quantized weights $w_1, w_2$ and their respective quantized inputs $x_1, x_2$. The standard accumulation would be performed as follows $w_1 x_1 + w_2 x_2$. In the case of PowerQuant, this would be done as $(w_1 x_1)^{\frac{1}{a}} + (w_2 x_2)^{\frac{1}{a}}$. Previous studies on quantization have demonstrated that such power functions can be computed with very high fidelity at almost no latency cost Kim et al. (2021).

## G    Overhead Cost of Zero-Points in Activation Quantization

The overhead cost introduced in equation 5 is well known in general in quantization as it arises from asymmetric quantization. Nonetheless, we share here (as well as in the article) some empirical values.

Table 8: Overhead induced by asymmetric quantization

| Architecture | parameters overhead | run-time overhead (CPU intel-m3) |
|---|---|---|
| ResNet50 | 0.25% | 4.35% |
| EfficientNet | 0.20% | 3.38% |
| ViT b16 | 0.73% | 5.14% |

Table 9: Comparison between the per-layer and global method of power parameter $a$ fitting on a ResNet 5à trained for ImageNet classification task.

| Architecture | Method | W-bit | A-bit | Accuracy | Reconstruction Error |
|---|---|---|---|---|---|
| | Baseline | 32 | 32 | 76.15 | - |
| | per-layer | 8 | 8 | 76.14 | **0.8** $\times 10^{-4}$ |
| ResNet 50 | global | 8 | 8 | **76.15** | 1.0 $\times 10^{-4}$ |
| | per-layer | 4 | 4 | 64,19 | **1.7** $\times 10^{-3}$ |
| | global | 4 | 4 | **70.29** | 1.9 $\times 10^{-3}$ |

These are empirical results from our own implementation. We include ResNet50 as it can also be quantized using asymmetric quantization although in our research, we only applied asymmetric quantization to SilU and GeLU based architectures. We included these results in the appendix of the revised article. It is worth noting that according to LSQ+ Bhalgat et al. (2020), asymmetric quantization can be achieved at virtually not run-time cost.

## H  LIMITATIONS OF THE RECONSTRUCTION ERROR METRIC

In the proposed PowerQuant method, we fit the parameter $a$ based on the reconstruction error over all the weights, *i.e.* over all layers in the whole network. Then, we perform per-channel quantization layer by layer independently. However, if the final objective is to minimize the reconstruction error from equation (3), a more efficient approach would consist in fitting the parameter $a$ separately for each layer. We note $a_l^*$ such that for every layer $l$ we have

$$a_l^* = \arg\min_a \left\{ \left\| W_l - Q_a^{-1}(Q_a(W_l)) \right\|_p \right\} \tag{21}$$

Then the network $(F, (a_l)^*)$ quantized with a per-layer fit of the power parameter will satisfy

$$\sum_{l=1}^L \left\| W_l - Q_{a_l}^{-1}(Q_{a_l}(W_l)) \right\|_p < \sum_{l=1}^L \left\| W_l - Q_a^{-1}(Q_a(W_l)) \right\|_p \tag{22}$$

if and only if their exists at least one $l$ such that $a_l \neq a$. Consequently, if the reconstruction error was a perfect estimate of the resulting accuracy, the per-layer strategy would offer an even higher accuracy than the proposed PowerQuant method. Unfortunately, the empirical evidence, in table 9, shows that the proposed PowerQuant method achieves better results in every benchmark. This observation demonstrates the limits of the measure of the reconstruction error. We explain this phenomenon by the importance of inputs and activations quantization. This can be seen as some form of overfitting the parameters $a_l$ on the weights which leads to poor performance on the activation quantization and prediction. In the general sens, this highlights the limitations of the reconstruction error as a proxy for maximizing the accuracy. Previous results can be interpreted in a similar way. For instance, in SQuant Cong et al. (2022) the author claim that it is better to minimize the absolute sum of errors rather than the sum of absolute errors and achieve good performance in data-free quantization.

## I  IMPROVEMENT WITH RESPECT TO QAT

In the introduction, we argued that data-driven quantization schemes performance define an upperbound on data-free performance. Our goal was to narrow the resulting gap between these methods. In Table 10, we report the evolution in the gap between data-free and data-driven quantization techniques. These empirical results validate the significant improvement of the proposed method at narrowing the gap between data-free and data-driven quantization methods by 26.66% to 29.74%.

Table 10: Performance Gap as compared to Data-driven techniques on ResNet 50 quantization in W4/A4. The relative gap improvement to the state-of-the-art SQuant [6], is measured as $\frac{gs-gp}{gs}$ with $gs = \frac{*-SQuant}{*}$ and $gp = \frac{*-PowerQuant}{*}$ where $*$ is the performance of a data-driven method

| data-driven method | SQuant | PowerQuant | relative gap |
|---|---|---|---|
| OCTAV Sakr et al. (2022) (ICML) | 8,72% | **6,15%** | +29,47% |
| SQ van Baalen et al. (2022) (CVPR) | 8,64% | **6,07%** | +29,74% |
| WinogradQ Chikin & Kryzhanovskiy (2022) (CVPR) | 9,55% | **7,00%** | +26,66% |
| Mr BiQ Jeon et al. (2022) (CVPR) | 8,74% | **6,17%** | +29,38% |

Table 11: Performance gap between data-free PowerQuant and short-retraining OCTAV Sakr et al. (2022).

| method | architecture | quantization | accuracy |
|---|---|---|---|
| PowerQuant | ResNet 50 | W4/A4 | 70.53 |
| OCTAV | ResNet 50 | W4/A4 | **75.84** |
| PowerQuant | MobileNet V2 | W4/A4 | **45.84** |
| OCTAV | MobileNet V2 | W4/A4 | 0.66 |

Table 12: Performance gap between PowerQuant and OCTAV Sakr et al. (2022) (using an additional short retraining), both using dynamic range estimation.

| method | architecture | quantization | accuracy |
|---|---|---|---|
| PowerQuant | ResNet 50 | W4/A4 | 76.02 |
| OCTAV | ResNet 50 | W4/A4 | **76.46** |
| PowerQuant | MobileNet V2 | W4/A4 | **71.65** |
| OCTAV | MobileNet V2 | W4/A4 | 71.23 |

In order to complete our comparison to QAT methods, we considered the short-re-training (30 epochs) regime from OCTAV in Table 11. We can draw two observations from this comparison. First, on ResNet 50, OCTAV achieves remarkable results by reach near full-precision accuracy. Still the proposed method does not fall too far back with only 5.31 points lower accuracy while being data-free. Second, on very small models such as MobileNet V2, using a strong quantization operator rather than a short re-training leads to a huge accuracy improvement as PowerQuant achieves 45.18 points higher accuracy. This is also the finding of the author in OCTAV, as they conclude that models such as MobileNet tend to be very challenging to quantize using static quantization and short re-training.

In Table 12, we draw a comparison between the proposed PowerQuant and the QAT method OCTAV Sakr et al. (2022), both using dynamic quantization (*i.e.* estimating the ranges of the activations on-the-fly depending on the input). As expected, the use of dynamic ranges has a considerable influence on the performance of both quantization methods. As can be observed the QAT method OCTAV achieved very impressive results and even outperforming the full-precision model on ResNet 50. Nevertheless, it is on MobileNet that the influence of dynamic ranges is the most impressive. For OCTAV, we observe a boost of almost 71 points going from almost random predictions to near exact full-precision accuracy. It is to be noted that PowerQuant does not fall shy in front of these performances, as using static quantization we still manage to preserve some of the predictive capability of the model. Furthermore, using dynamic quantization, Powerquant achieves similar accuracies than OCTAV while not involving any fine-tuning, contrary to OCTAV.

All in all, we can conclude that the proposed data-free method manages to hold close results to a state-of-the-art QAT method in some context. An interesting future work could be the extension of PowerQuant as a QAT method and possibly learning the power parameter $a$ that we use in our quantization operator.

## J    COMPARISON TO STATE-OF-THE-ART DATA-FREE QUANTIZATION ON OTHER CONVNETS

In addition to our evaluation on ResNet, we propose some complementary results on DenseNet in Table 13 as well as the challenging and compact architectures MobileNet and EfficientNet in Table

Table 13: Comparison between state-of-the-art post-training quantization techniques on DenseNet 121 on ImageNet. We distinguish methods relying on data (synthetic or real) or not. In addition to being fully data-free, our approach significantly outperforms existing methods.

| Architecture | Method | Data | W-bit | A-bit | Accuracy | gap |
|---|---|---|---|---|---|---|
| | Baseline | - | 32 | 32 | 75.00 | - |
| | DFQ Nagel et al. (2019) | No | 8 | 8 | 74.75 | -0.25 |
| | SQuant Cong et al. (2022) | No | 8 | 8 | 74.70 | -0.30 |
| | OMSE Choukroun et al. (2019) | Real | 8 | 8 | 74.97 | -0.03 |
| | SPIQ Yvinec et al. (2022b) | No | 8 | 8 | **75.00** | **-0.00** |
| DenseNet 121 | PowerQuant | No | 8 | 8 | **75.00** | **-0.00** |
| | DFQ Nagel et al. (2019) | No | 4 | 4 | 0.10 | -74.90 |
| | SQuant Cong et al. (2022) | No | 4 | 4 | 47.14 | -27.86 |
| | SPIQ Yvinec et al. (2022b) | No | 4 | 4 | 51.83 | -23.17 |
| | OMSE Choukroun et al. (2019) | Real | 4 | 4 | 57.07 | -17.93 |
| | PowerQuant | No | 4 | 4 | **69.37** | **-5.63** |

Table 14: Complementary Benchmarks on ImageNet

| Architecture | Method | Data | W-bit | A-bit | Accuracy | gap |
|---|---|---|---|---|---|---|
| | Baseline | - | 32 | 32 | 71.80 | - |
| | DFQ (ICCV 2019) | No | 8 | 8 | 70.92 | -0.88 |
| | SQuant (ICLR 2022) | No | 8 | 8 | 71.68 | -0.12 |
| | SPIQ (WACV 2023) | No | 8 | 8 | 71.79 | -0.01 |
| MobileNet V2 | PowerQuant | No | 8 | 8 | **71.81** | **+0.01** |
| | DFQ (ICCV 2019) | No | 4 | 4 | 27.1 | -44.70 |
| | SQuant (ICLR 2022) | No | 4 | 4 | 28.21 | -43.59 |
| | SPIQ (WACV 2023) | No | 4 | 4 | 31.28 | -40.52 |
| | PowerQuant | No | 4 | 4 | **45.84** | **25.96** |
| | Baseline | - | 32 | 32 | 77.10 | - |
| | DFQ (ICCV 2019) | No | 8 | 8 | 76.89 | -0.21 |
| | SQuant (ICLR 2022) | No | 8 | 8 | 76.93 | -0.17 |
| | SPIQ (WACV 2023) | No | 8 | 8 | 77.02 | -0.08 |
| EfficientNet B0 | PowerQuant | No | 8 | 8 | **77.05** | **-0.05** |
| | DFQ (ICCV 2019) | No | 6 | 6 | 43.08 | -34.02 |
| | SQuant (ICLR 2022) | No | 6 | 6 | 54.51 | -32.59 |
| | SPIQ (WACV 2023) | No | 6 | 6 | 74.67 | -2.43 |
| | PowerQuant | No | 6 | 6 | **75.13** | **-1.97** |

14 as well as weights only for Bert in Table 16. In table 13, we report the performance of other data-free quantization processes on DenseNet 121. The OMSE method (Choukroun et al., 2019) is a post-training quantization method that leverages validation examples during quantization, thus cannot be labelled as data-free. Yet, we include this work in our comparison as they show strong performance in terms of accuracy at a very low usage of real data. As showcased in table 13, the proposed PowerQuant method almost preserves the floating point accuracy in W8/A8 quantization. Additionally, on the challenging W4/A4 setup, our approach improves the accuracy by a remarkable 12.30 points over OMSE and 17.54 points over SQuant. This is due to the overall better efficiency of non-uniform quantization, that allows a theoretically closer fit to the weight distributions of each DNN layer. The results on MobileNet and EfficientNet from Table 14 confirm our previous findings. We observe a significant boost in performance from PowerQuant as compared to the other very competitive data-free solutions.

## K  OVERHEAD COST DISCUSSION

In this section, we provide more empirical results on the inference cost of the proposed method. Table 17 shows the inference time of DNNs quantized with our approach (which only implies modifications of the activation function and a bias correction-see Section 3.3). For DenseNet, ResNet and MobileNet V2, the baseline activation function is the ReLU, which is particularly fast to compute.

Table 15: Complementary Benchmarks on Vision Transformers for ImageNet

| Architecture | Method | Data | W-bit | A-bit | Accuracy | gap |
|---|---|---|---|---|---|---|
| | Baseline | - | 32 | 32 | 78.524 | - |
| | DFQ (ICCV 2019) | No | 8 | 8 | 77.612 | -0.912 |
| | SQuant (ICLR 2022) | No | 8 | 8 | 77.638 | -0.886 |
| CaiT xxs24 | PowerQuant | No | 8 | 8 | **77.718** | **-0.806** |
| | DFQ (ICCV 2019) | No | 4 | 8 | 74.192 | -4.332 |
| | SQuant (ICLR 2022) | No | 4 | 8 | 74.224 | -4.300 |
| | PowerQuant | No | 4 | 8 | **75.104** | **-3.420** |
| | Baseline | - | 32 | 32 | 79.760 | - |
| | DFQ (ICCV 2019) | No | 8 | 8 | 79.000 | -0.760 |
| | SQuant (ICLR 2022) | No | 8 | 8 | 78.914 | -0.846 |
| CaiT xxs36 | PowerQuant | No | 8 | 8 | **79.150** | **-0.610** |
| | DFQ (ICCV 2019) | No | 4 | 8 | 76.906 | -2.854 |
| | SQuant (ICLR 2022) | No | 4 | 8 | 76.896 | -2.864 |
| | PowerQuant | No | 4 | 8 | **77.702** | **-2.058** |
| | Baseline | - | 32 | 32 | 83.368 | - |
| | DFQ (ICCV 2019) | No | 8 | 8 | **82.802** | **-0.566** |
| | SQuant (ICLR 2022) | No | 8 | 8 | 82.784 | -0.584 |
| CaiT s24 | PowerQuant | No | 8 | 8 | 82.766 | -0.602 |
| | DFQ (ICCV 2019) | No | 4 | 8 | 81.474 | -1.894 |
| | SQuant (ICLR 2022) | No | 4 | 8 | 81.486 | -1.882 |
| | PowerQuant | No | 4 | 8 | **81.612** | **-1.756** |

Table 16: Complementary Benchmarks on the GLUE task. We consider the BERT transformer architecture. We provide the reference performance of BERT on GLUE as well as our reproduced results (baseline).

| task | (reference) | baseline | uniform | log | power |
|---|---|---|---|---|---|
| CoLA | 49.23 | 47.90 | 46.24 | 46.98 | **47.77** |
| SST-2 | 91.97 | 92.32 | 91.28 | 91.85 | **92.32** |
| MRPC | 89.47/85.29 | 89.32/85.41 | 86.49/81.37 | 86.65/82.86 | **89.32/85.41** |
| STS-B | 83.95/83.70 | 84.01/83.87 | 83.25/83.14 | **84.01**/83.81 | **84.01/83.87** |
| QQP | 88.40/84.31 | 90.77/84.65 | 90.23/84.61 | 90.76/**84.65** | **90.77/84.65** |
| MNLI | 80.61/81.08 | 80.54/80.71 | 79.72/79.13 | 79.22/79.71 | **80.54/80.71** |
| QNLI | 87.46 | 91.47 | 90.32 | 91.43 | **91.47** |
| RTE | 61.73 | 61.82 | 59.23 | 61.27 | **61.68** |
| WNLI | 45.07 | 43.76 | 40.85 | 42.80 | **42.85** |

Nevertheless, our results show that our approach leads to only increasing by $1\%$ the whole inference time on most networks. More precisely, in the case of ResNet 50, the change in activation function induces a slowdown of $0.15\%$. The largest runtime increase is obtained on DenseNet with a 3.4% increase. Lastly, note that our approach is also particularly fast and efficient on EfficientNet B0, which uses SiLU activation, thanks to the bias correction technique introduced in Section 3.3. Overall, the proposed approach can be easily implemented and induces negligible overhead in inference on GPU. To furthermore justify the practicality of the proposed quantization process, we recall that the only practicality concern that may arise is on the activation function as the other operations are strictly identical to standard uniform quantization. According to Kim et al. (2021) efficient power functions can be implemented for generic hardware as long as they support standard integer arithmetic, i.e. as long as they support uniform quantization. When it comes to Field-Programmable Gate Array (FPGA), activation functions are implemented using look-up tables (LUT) as detailed in Hajduk (2017). More precisely, they are pre-computed using Padé approximation which are quotients of polynomial functions. Consequently the proposed approach would simply change the polynomial values but not the inference time as it would still rely on the same number of LUTs.

In general, activation functions that are non-linear can be very effectively implemented in quantization runtime Lam et al. (2022). However these considerations are hardware agnostic. In order to circumvent this limitation and address any concerns to our best, we conducted a small study using

Table 17: Inference time, in seconds, over ImageNet using batches of size 16 of several networks on a 2070 RTX GPU. We also report the accuracy for W6/A6 quantization setup.

| Architecture | Method | inference time (gap) | accuracy |
|---|---|---|---|
| ResNet 50 | Uniform | 164 | 75.07 |
| | Power Function | 164 (+0.2) | 75.95 |
| DenseNet 121 | Uniform | 162 | 72.71 |
| | Power Function | 167 (+4.8) | 74.84 |
| MobileNet V2 | Uniform | 85 | 52.20 |
| | Power Function | 86 (+0.7) | 64.09 |
| EfficientNet B0 | Uniform | 125 | 58.24 |
| | Power Function | 127 (+2.2) | 66.38 |

Table 18: Inference cost each component of a convolutional layer and percentage of total in terms of number of cycles on a wide range of simulated hardware using nntool from GreenWaves.

| operation | number of cycles | number of ops | % of total cycles | % of total ops |
|---|---|---|---|---|
| convolution | 22950 | 442368 | 85.482% | 99.310% |
| bias | 2033 | 1024 | 7.573% | 0.229% |
| relu | 924 | 1024 | 3.442% | 0.229% |
| power function | 940 | 1024 | 3.502% | 0.229% |

Table 19: We report the processing time in seconds (on an Intel(R) Core(TM) i9-9900K CPU) required to quantize a trained neural network such as ResNet, MobileNet, DenseNet or EfficientNet.

| Architecture | GDFQ | SQuant | Uniform | Power |
|---|---|---|---|---|
| MobileNet V2 | $7.10^3$ | 134 | <**1** | <**1** |
| ResNet 50 | $11.10^3$ | 320 | <**1** | 1.3 |

the simulation tool nntool from GreenWaves, a risc-v chips manufacturer that enables to simulate inference cost of quantized neural networks on their gap unit. We tested a single convolutional layer with bias and relu activation plus our power quantization operation and reported the number of cycles and operations. These results demonstrate that even without any optimization the proposed method has a marginal computational cost on MCU inference which corroborates our previous empirical results. We would like to put the emphasis on the fact that this cost could be further reduced via optimizing the computation of the power function using existing methods such as Kim et al. (2021). Similarly, we measure the empirical time required to perform the proposed quantization method on several neural networks and report the results in table 19. These results show that the proposed PowerQuant method offers outstanding trade-offs in terms of compression and accuracy at virtually no cost over the processing and inference time as compared to other data-free quantization methods. For instance, SQuant is a sophisticated method that requires heavy lifting in order to efficiently process a neural network. On a CPU, it requires at least 100 times more time to reach a lower accuracy than the proposed method as we will showcase in our comparison to state-of-the-art quantization schemes.

