# OpenReview forum: "PowerQuant: Automorphism Search for Non-Uniform Quantization"
_ICLR.cc/2023/Conference — ICLR 2023 poster_

### Official Review · Reviewer_TJBp · 2022-10-21

**Confidence:** 4
**Correctness:** 3
**Technical Novelty And Significance:** 3
**Empirical Novelty And Significance:** 2
**Recommendation:** 6

**Clarity, Quality, Novelty And Reproducibility:**

I understand this field is moving very fast and it is hard to keep track to the most recent related works. But most of the works that were compared to are already dated by a few years. Can the authors compare to more recent works such as [1]

[1] Sakr, Charbel, et al. "Optimal Clipping and Magnitude-aware Differentiation for Improved Quantization-aware Training." International Conference on Machine Learning. PMLR, 2022.

**Strength And Weaknesses:**

Strengths: - The paper proposes a novel quantization technique based on power functions. - Some theoretical analysis is performed to describe the method. - Extensive experiments are performed to evaluate the method. Weaknesses: - The mathematical derivations are hard to follow. I think there is room for improvement in terms of clarity.

**Summary Of The Paper:**

This paper looks at data free techniques for quantization. The framework is to look for automorphisms on the (R,x) which the authors show is essentially power functions. Then exponential representation are optimized to minimize reconstruction error. Finally, experimental results are presented to show the usefulness of the work.

**Summary Of The Review:**

The paper proposes a new quantization method, provides analyses and empirical evaluations.

During the rebuttal and through discussions with the authors, I have raised my score to a 6.

---

> ### Author Response · Authors · 2022-11-06
> **Response to reviewer TJBp**
>
> **Concern on Clarity:** Thank you for your feedback. To answer your concerns about presentation clarity, we reworked the methodology sections to more progressively introduce those ideas and the intuition behind them, and to provide more insight regarding the mathematical derivations. We hope that these changes in the revised version address your issues on paper understandability.
>
> **Comparison to other methods:** Although we did provide comparisons with OCTAV [1] in Appendix I of the paper (see Table 10), we added more details to the existing comparisons (see Table 11 of the revised manuscript and Table below). However, please keep in mind that OCTAV [1] is not data-free and involves re-training which is why we classified it as a quantization aware training method (QAT) while our method is applied post-training (PTQ) and is data-free.
>
> | method | architecture | quantization | accuracy |
> | --- | --- | --- | ---|
> | PowerQuant | ResNet 50 | W4/A4 | 70.53 |
> | OCTAV | ResNet 50 | W4/A4 | **75.84** |
> | PowerQuant | MobileNet V2 | W4/A4 | **45.84** |
> | OCTAV | MobileNet V2 | W4/A4 | 0.66 |
>
> Indeed, authors in [1] use 30 epochs for re-training while powerQuant is fully data-free. We can draw two observations from this comparison. First, on ResNet 50, OCTAV achieves remarkable results and reach near full-precision accuracy. Still, the proposed method does not fall too far behind with only 5.31 points lower accuracy while being data-free. Second, on very small models such as MobileNet V2, using a strong quantization operator rather than a short re-training leads to a huge accuracy improvement as PowerQuant achieves 45.18 points higher accuracy. This is also the finding of the author in OCTAV, as they conclude that models such as MobileNet tend to be very challenging to quantize using static quantization and short re-training. We believe that this comparison shows the strength of the proposed approach, and included this discussion in the revised version of the article, in appendix I.
>
> **References**
>
> [1] Sakr, Charbel, et al. "Optimal Clipping and Magnitude-aware Differentiation for Improved Quantization-aware Training." International Conference on Machine Learning. PMLR, 2022.

---

> > ### Comment · Reviewer_TJBp · 2022-11-06
> > **Thanks for the response**
> >
> > Thank you for the response. It should be noted that with additional training time, 4-bit ResNet-50 accuracy reaches **76.46%** and that with a dynamic quantizer, 4-bit MobileNet-V2 accuracy reaches **71.23%**. These are listed in Table 4 of the OCTAV paper. While experimental setups might slightly differ, these results do provide an estimate for the potential of the achievable accuracy with 4-bit. The authors should mention these and provide any comparisons and comments.

---

> > > ### Author Response · Authors · 2022-11-06
> > > **Response to reviewer TJBp**
> > >
> > > We would like to thank the reviewer on their precision about their concern and their valuable advice. To answer your concerns about the comparison with recent QAT method OCTAV, we ran benchmarks in a closer setup and added them to the revised article. We list these results in the table below:
> > >
> > > | method | architecture | range | fine-tuning | quantization | accuracy |
> > > | --- | --- | --- | ---| --- | ---|
> > > | OCTAV | ResNet 50 | static | 30 epochs | W4/A4 | 75.84 |
> > > | OCTAV | ResNet 50 | dynamic | 30 epochs| W4/A4 | **76.46** |
> > > | PowerQuant | ResNet 50 | static | None | W4/A4 | 70.53 |
> > > | PowerQuant | ResNet 50 | dynamic | None | W4/A4 | 76.02 |
> > > | OCTAV | MobileNet V2 | static | 30 epochs | W4/A4 | 0.66 |
> > > | OCTAV | MobileNet V2 | dynamic | 30 epochs | W4/A4 | 71.23 |
> > > | PowerQuant | MobileNet V2 | static | None | W4/A4 | 45.84 |
> > > | PowerQuant | MobileNet V2 | dynamic | None | W4/A4 | **71.65** |
> > >
> > > As you pointed out, the use of additional training time and dynamic range estimation has a considerable influence on the performance of the quantization method. On the one hand, OCTAV achieve very impressive results in this setup, even outperforming the full-precision model on ResNet 50. The proposed PowerQuant, however, achieves very close performance using dynamic range estimation, almost reaching exact full-precision accuracy despite not requiring any fine-tuning of the network.
> > >
> > > On MobileNet V2 that the influence of dynamic ranges is even more impressive: for OCTAV, the accuracy goes from almost chance level to near full-precision accuracy. On the other hand, PowerQuant does not fall shy in front of this performance, as using static quantization we still manage to preserve some of the predictive capability of the model. Lastly, using dynamic quantization, PowerQuant achieves a remarkable 71.65% accuracy.
> > >
> > > We added those results to Appendix I of the revised version. We believe that these results further shows the interest of the proposer power quantization approach, as well as the benefits of the proposed ideas and results for the community, and hope that it answers your concerns as well.

---

> > > > ### Comment · Reviewer_TJBp · 2022-11-07
> > > > **Sounds good**
> > > >
> > > > Sounds good, please be sure to include a summary of these findings in the main paper. I update the score to a 6.

---

> > > > > ### Author Response · Authors · 2022-11-08
> > > > > **Main paper update**
> > > > >
> > > > > We updated the revised article and included the discussed results in section 4.3

---

### Official Review · Reviewer_6h14 · 2022-10-25

**Confidence:** 4
**Clarity, Quality, Novelty And Reproducibility:** see above
**Correctness:** 4
**Technical Novelty And Significance:** 3
**Empirical Novelty And Significance:** 3
**Recommendation:** 6

**Strength And Weaknesses:**

### Strength
* This paper is well-motivated. Bell-shaped distribution of weights is widely observed. Power transformation addresses this issue well.
* Continuous automorphisms are highly appreciated by quantization and hardware implementation. This method leverages continuous automorphisms in a smart way.
* Extensive experiments are conducted on various architectures and datasets with significant improvements.


### Weaknesses
* Is $W$ in Eq. 2 a matrix? What is the precise definition of $|\cdot|^a$?
* "for instance, for both SiLU (EfficientNets) and GeLU (Image Transformers), the activations are bounded below" Are they empirical observations? How many examples are tested?
* Although the authors mentioned the overhead of power transformation (against uniform quantization) in Eq (5), it should be further analyzed in detail to help understand the trade-off.
* Missing reference on power-based quantization [1] [2]

[1] Zhang, Sai Qian, et al. "Training for multi-resolution inference using reusable quantization terms." Proceedings of the 26th ACM International Conference on Architectural Support for Programming Languages and Operating Systems. 2021.
[2] Li, Yuhang, Xin Dong, and Wei Wang. "Additive powers-of-two quantization: An efficient non-uniform discretization for neural networks." ICLR (2020).

**Summary Of The Paper:**

This paper proposes to transform weights with power operation before quantization to improve the quantized accuracy. The motivation behind this is the bell-shaped distribution of weights, indicating that fine-grained resolution should be allocated around zero. Extensive experiments are conducted to demonstrate the effectiveness of the proposed method against several strong baselines.

**Summary Of The Review:**

Both methods and results are interesting and significant. I am looking forward to feedback from the authors on my questions.

---

> ### Author Response · Authors · 2022-11-06
> **Response to reviewer 6h14**
>
> **Mathematical Notations:** In equation (2), $W$ is a weight tensor, which corresponds to a matrix in the case of fully-connected layers and a four dimensional tensor in the case of convolutional layers. All the operations applied to $W$ are element-wise operations, which means that, in the case of a fully-connected layer.
>
> $$
> |W|^a = \left|\left(\begin{matrix} W_{1,1} & ... & W_{1,n} \\\\ \vdots & \ddots & \vdots \\\\ W_{m,1} & ... & W_{m,n} \end{matrix}\right)\right|^a = \left(\begin{matrix} |W_{1,1}|^a & ... & |W_{1,n}|^a \\\\ \vdots & \ddots & \vdots \\\\ |W_{m,1}|^a & ... & |W_{m,n}|^a\end{matrix}\right)
> $$
> We explained this in Section 3.1 of the revised paper to improve clarity.
>
> **Lower bounds on SiLU and GeLU:** Lower bounds on the functions are mathematically derived. This result does not come from an empirical evaluation. This is a very important detail and thank the reviewer for pointing it out. We edited the submission in order to clarify this point. As we need to estimate ranges in order to quantize the activations, knowing that the lower bound is mathematically exact is of paramount importance.
>
> **Overhead computation induced by equation (5):** The overhead cost introduced in equation 5 is well known in general in quantization as it arises from asymmetric quantization. In general, asymmetric quantization allows one to have a better coverage of the quantized values support. In our case, we use asymmetric quantization to work with positive values only. Nonetheless, we share here (as well as in the article) some empirical values.
>
> | model | overhead #params | overhead run-time |
> | --- | --- | --- |
> | ResNet50 | 0.25% | 4.35% |
> | EfficientNet | 0.20% | 3.38% |
> | ViT b16 | 0.73% | 5.14% |
>
> These are empirical results from our own implementation. We include ResNet50 as it can also be quantized using asymmetric quantization although in our research, we only applied asymmetric quantization to SilU and GeLU based architectures. We included these results in the appendix of the revised article. It is worth noting that according to LSQ+ [1], asymmetric quantization can be achieved at virtually not run-time cost.
>
> **Missing references:** The reviewer pointed-out two major articles in non-uniform quantization [2,3]. We added them to the revised version. Here, we will discuss how these articles relate, but differ from our method. First and foremost, unlike PowerQuant, these methods are data-driven as they use data in order to quantize efficiently the model. Second, like PowerQuant, these methods are non-uniform, however, they use log quantization with an arbitrary basis ($a^X$), while we use power function quantization ($X^a$). We thank the reviewer for sharing these important references that we added to the related work section.
>
> **References**
>
> [1] Bhalgat Yash, Lee Jinwon, Nagel Markus, Blankevoort Tijmen and Kwak Nojun. Lsq+: Improving low-bit quantization through learnable offsets and better initialization. CVPR Workshops}, pp. 696--697, 2020.
>
> [2] Zhang, Sai Qian, et al. "Training for multi-resolution inference using reusable quantization terms." Proceedings of the 26th ACM International Conference on Architectural Support for Programming Languages and Operating Systems. 2021.
>
> [3] Li, Yuhang, Xin Dong, and Wei Wang. "Additive powers-of-two quantization: An efficient non-uniform discretization for neural networks." ICLR (2020).

---

> > ### Comment · Reviewer_6h14 · 2022-11-15
> > **Thanks for the response**
> >
> > The authors answer my clarification questions well. Thanks a lot.
> > In addition, for activation quantization, I recommend the authors can provide some insights for readers instead of merely pointing out the reference of I-bert.

---

> > > ### Author Response · Authors · 2022-11-15
> > > **Response to reviewer 6h14**
> > >
> > > Thank you for your response. As suggested, we added those insights in section 3.3 of the revised manuscript. We hope that these clarifications answer your last concerns.

---

### Official Review · Reviewer_Htvb · 2022-10-27

**Confidence:** 3
**Clarity, Quality, Novelty And Reproducibility:** Please read "Strength And Weaknesses"…
**Correctness:** 3
**Technical Novelty And Significance:** 4
**Empirical Novelty And Significance:** 4
**Recommendation:** 6

**Strength And Weaknesses:**

Strength:

- Novelty
- The claims are backed by mathematical explanation
- promising results

Weaknesses:

- The paper is heavy and very hard to read. To someone like me who is a non-native English speaker, the paper looks like a poorly translated novel that consists of unnecessary fancy words and grammars. It was hard to keep reading the paper and not getting distracted.
- Section "3.3 FUSED DE-QUANTIZATION AND ACTIVATION FUNCTION" needs more clarification. I could not understand how the quantization works for the activations. Do we calculate $x^{\alpha}$ online during the inference for the activations? If yes (and I think the answer is yes) then how is it calculated?

**Summary Of The Paper:**

The paper proposed a novel non-uniform data-free quantization method.
The main Idea (which is very interesting) is to find the best non-uniform quantization intervals for both weights and activations by minimizing the reconstruction error using the proposed quantization and de-quantization method.
The paper provides extensive experimentation on various datasets and tasks (including Image-classification and NLP) and archives promising results (outperforms  SOTA).

**Summary Of The Review:**

As mentioned in the Strength And Weaknesses section, I did not understand the quantization of activations. Hence, I need to rely on the answer from the authors and other reviewers comments. I am open to change my score in future.

---

> ### Author Response · Authors · 2022-11-06
> **Response to reviewer Htvb**
>
> **Readability:** The main concern on readability was the unnecessary use of fancy words and grammars. In particular in the methodology, we tried to present the intuition behind why searching for non-uniform quantization operators led to search for the automorphism of $(\mathbb{R}^*_+, \times)$, which turn out to be power functions. To answer your concerns, we reworked the methodological sections to more clearly and progressively introduce those ideas and the intuition behind them, as well as to explain more the mathematical derivations. We hope that these changes address your concerns for improved readability.
>
> **Quantization of the activations:** In order to quantize the activations, we have to compute a power of $x$. In most instances, we compute $x^a = \sqrt{x}$. This can be efficiently done in full integer arithmetic as discussed in I-Bert [1]. Their algorithm uses Newton's method and converges in 2 steps for low bit representations (four steps for int32). They demonstrate the runtime efficiency of the method which is why we relied on it. In our original submission, the activation quantization implementation details were thoroughly discussed in Appendix F. However, as it was pointed out in the review as a concern, we added discussion on it in Section 3.3.
>
> **Reférences**
> [1] Sehoon Kim, Amir Gholami, Zhewei Yao, Michael W Mahoney, and Kurt Keutzer. I-bert: Integeronly bert quantization. In ICML, pp. 5506–5518. PMLR, 2021.

---

> > ### Public Comment · ~Andrew_Stevens2 · 2023-03-01
> > **Accuracy results and claimed efficiency not realizeable**
> >
> > The claimed 'headline' accuracies rely on optimal choices of a != 1/2  However the efficient Newton-Raphson approximation of x^a* relies on a* == 1/2.
> >
> > Performance on the ResNet is markedly worse for this choice of a*.
> >
> > Similarly as becomes clear only in  Appendix F Algorithm 2 is grossly misleading.
> > It is NOT possible, as this seems to imply, to compute neural networks layer sum-of-products calculations 'normally'  using this quantized representation. Instead of summing  (narrow bitwidth) products it is necessary to sum roots of products.  Even for a*==1/2 the root operation is itself of course an order of magnitude more complex than a simple product requiring multiple significantly  wider multiplications and additions.
> >
> > The proposed method is thus not, as claimed, realizeable in practical HW or SW implementations with 'negligble overhead'.

---

> > > ### Author Response · Authors · 2023-03-02
> > > **Response de Andrew Stevens**
> > >
> > > Hi Andrew,
> > >
> > > First and foremost, although the performance decreases for a=0.5 on ResNet it remains largely above other data-free techniques. So the claimed headline accuracy relies on a choice of a close to $a^*$ which allows to achieve rely good results using a=0.5. For instance, all of our results on transformers are obtained without using $a^*$ but rather a=0.5. For instance, with $a=0.5$, on ViT, we can reach above previous 8 bits sota accuracy, using only 4 bits on the weights.
> > >
> > > You state that `the root operation is itself of course an order of magnitude more complex than a simple product requiring multiple significantly wider multiplications and additions`. However, it is the opposite, the square root does not increase the required number of bits as the square root $x$ fits on less bits than $x$... For a more thorough discussion on inference overhead please see Appendix K.
> > > If you found algorithm 2 misleading, we can edit it to clarify that the square root is implied on the accumulation although it is clearly stated in the text.

---

### Decision · Program_Chairs · 2023-01-20

**Decision:**

Accept: poster

**Justification For Why Not Higher Score:**

Its contribution is not strong enough for a spoylight paper. The writing is bad.

**Justification For Why Not Lower Score:**

All reviewers agreed that this work is novel and mathematically rigorous. Moreover, the extensive experiments show the good performance of the proposed method.

**Metareview: Summary, Strengths And Weaknesses:**

This paper studies data-free non-uniform quantization for deep neural networks, which can help lower the number of bits that are used for encoding weights and activations. The framework is to search among automorphisms which are essentially power functions. All reviewers agreed that this work is novel and mathematically rigorous. Moreover, the extensive experiments show the good performance of the proposed method. That being said, most reviewers complained that this work was poorly written and hard to follow. I strongly suggest that the authors polish this paper in the revised version.  I suggest acceptance.

**Note From Pc:**

if the above contains the word "oral" or "spotlight" please see: "oral" presentation means -> notable-top-5% and "spotlight" means -> notable-top-25%. As stated in our emails, we are disassociating presentation type from AC recommendations